# Tumor Cell-Specific Retention and Photodynamic Action of Erlotinib-Pyropheophorbide Conjugates

**DOI:** 10.3390/ijms231911081

**Published:** 2022-09-21

**Authors:** Erin C. Tracy, Ravindra R. Cheruku, Ravindra K. Pandey, Heinz Baumann

**Affiliations:** 1Department of Molecular & Cellular Biology, Roswell Park Comprehensive Cancer Center, Buffalo, NY 14263, USA; 2Department of Cell Stress Biology, Roswell Park Comprehensive Cancer Center, Buffalo, NY 14263, USA

**Keywords:** erlotinib-conjugates, cellular uptake, subcellular deposition, photoreaction, EGFR inhibition, light-sensitive erlotinib release

## Abstract

To enhance uptake of photosensitizers by epithelial tumor cells by targeting these to EGFR, pyropheophorbide derivatives were synthesized that had erlotinib attached to different positions on the macrocycle. Although the addition of erlotinib reduced cellular uptake, several compounds showed prolonged cellular retention and maintained photodynamic efficacy. The aim of this study was to identify whether erlotinib moiety assists in tumor targeting through interaction with EGFR and whether this interaction inhibits EGFR kinase activity. The activity of the conjugates was analyzed in primary cultures of human head and neck tumor cells with high-level expression of EGFR, and in human carcinomas grown as xenografts in mice. Uptake of erlotinib conjugates did not correlate with cellular expression of EGFR and none of the compounds exerted EGFR-inhibitory activity. One derivative with erlotinib at position 3, PS-10, displayed enhanced tumor cell-specific retention in mitochondria/ER and improved PDT efficacy in a subset of tumor cases. Moreover, upon treatment of the conjugates with therapeutic light, EGFR-inhibitory activity was recovered that attenuated EGFR signal-dependent tumor cell proliferation. This finding suggests that tumor cell-specific deposition of erlotinib-pyropheophorbides, followed by light triggered release of EGFR-inhibitory activity, may improve photodynamic therapy by attenuating tumor growth that is dependent on EGFR-derived signals.

## 1. Introduction

Past studies of carcinoma cells derived from surgical specimens of lung and head and neck (HN) tumor tissues have indicated a large range by which the lead second generation photosensitizer (PS), HPPH [3-(1′-hexyloxy)ethyl-3-devinyl pyropheophorbide-a] is taken up and retained by tumor cells [1,2]. Although tumor cells from ~50% of cases, after placing in primary cultures, showed a preferential retention of HPPH over co-resident stromal cells, tumor cells from other cases indicated substantially reduced retention. By applying variants of HPPH, which differ primarily in the presence of side groups on the tetrapyrrole macrocycle, structural elements for enhanced cancer cell-specific retention have been identified [3,4,5]. Most observed was the prolonged retention of compounds with neutralized charge at position 17^2^, as exemplified by methyl ester derivatives [5]. Not surprisingly, comparable methyl ester derivatives of near-infrared PSs have emerged as optimized compounds entering clinical trials [6,7].

To increase uptake of PSs by tumor cells, a targeted delivery was approached. The rationale was that, because epithelial cancer cells, such as derived from lung, HN, or bladder, have elevated expression of EGFR, uptake of PSs, including HPPH, could be improved by using EGFR-interacting forms [8,9]. The basic strategy was to modify HPPH and related chlorins at different peripheral positions by the addition of the small-molecular EGFR inhibitor erlotinib [10,11]. The initial functional evaluation was made with HPPH-related pyropheophorbides containing erlotinib conjugated to position 3^1^, 13, 17^2^, or 20 via a benzoic acid linker [11]. The attachment involved either a rigid or flexible, short or long linker structure. The biological assays of the various products were restricted to the quantification of cellular uptake in tissue culture and photodynamic action in vivo using established tumor models. The results identified one erlotinib conjugate, PS 10, which exerted a photodynamic therapy (PDT) efficacy exceeding that of HPPH.

Two interrelated questions remained to be answered concerning the functionality of the erlotinib-pyropheophorbide-a conjugates: (a) to what extent is the EGFR-binding and inhibitory activity of the erlotinib moiety retained and (b) does it contribute to the PDT outcome? We expected that the level of photoreaction mediated by the chlorin portion is not affected by the presence of erlotinib and is proportional to the amount of the PS present in tumor cells [11].

The current study determined the effectiveness of the chimeric photosensitizers to be taken up relative to the level of EGFR expression, to mediate a proportional level of photoreaction, and to modulate EGFR functions. The results demonstrate that erlotinib attached to pyropheophorbide, despite the predicted appropriate structural interaction with the target EGFR site, does not appreciably mediate EGFR-dependent uptake and does not inhibit EGFR kinase activity. However, upon treatment of a subset of erlotinib conjugates with therapeutic light, EGFR-inhibitory activity is recovered that is effective in attenuating EGFR kinase activity and EGFR signal-dependent tumor cell proliferation.

## 2. Results and Discussion

### 2.1. Erlotinib Conjugates and Cellular Uptake

The current functional study involved examples of erlotinib chlorin conjugates described previously [11] and which are depicted in Figure 1. The original designations of these compounds were retained to facilitate cross-comparison of data. Sets of erlotinib conjugates had been synthesized to evaluate the functional impact of several specific structural features: flexible or rigid linker (PS 5 vs. PS 8); attachment of erlotinib to position 17^2^ or position 3^1^ (PS 8 vs. PS 10); contribution of an acidic charge at position 17^2^ to cellular retention (PS 10 vs. PS 11); attachment of erlotinib to the meta or para position of the rigid benzyloxy linker (PS 10 vs. PS 16), a chiral methyl group at position 3^1^ of pyropheophorbide (PS 10 vs. PS 19); increase to three erlotinib moieties (PS 8 vs. PS 27); and attachment of erlotinib to position 13 of chlorin e6 with short or long linker structure (PS 38 vs. PS 43). The initial characterization of additional compounds had already ruled out as ineffective those PSs with erlotinib conjugated to position 20 via benzoic acid linker of pyropheophorbide (PS 23) or to the imide ring of purpurinimide (PS 29 and PS 31).

Following information regarding cellular activity had been obtained [11]. Erlotinib attached by rigid linker reduced cellular uptake. Uptake was further lowered when erlotinib was placed into para position of the benzyloxy linker. Retention was also attenuated when pyropheophorbide is missing the chiral methyl group at position 3^1^. The intracellular sites of deposition for every PS with a single erlotinib moiety appeared to be the same as for non-conjugated HPPH. The highest level of cellular uptake, but coupled with reduced tumor cell specificity, was determined for PS 5. In contrast, PS 10, although taken up at moderate level, demonstrated highest differential retention by a subset of tumor cells. Uptake and photodynamic activities measured in cell culture systems could be confirmed in vivo using xenografts of established human cancer cell lines. However, none of the preliminary cell studies provided evidence for a specific involvement of an erlotinib-sensitive and EGFR-dependent action. The current study addresses this issue.

To be effective as substrate for binding to EGFR, as well as to act as inhibitor of the EGFR kinase, erlotinib conjugates must gain access to the cytoplasmic domain of the EGFR protein. Hence, excluded are those compounds that are endocytosed and deposited to lysosomes, such as PS 27. To visualize the mode of cellular entry and intracellular localization of the erlotinib conjugates under investigation, we employed reconstituted co-cultures of head and neck (HN) tumor epithelial cells (T-EC) and tumor-associate fibroblasts (T-Fb) (Figure 2; example of tongue tumor-derived HNT1 cells combined with carboxyfluorescein succinimidyl ester (CFSE)-tagged HN tumor stromal cells). Incubations for 30 min demonstrated the immediate and high-level entry of HPPH by diffusion into both cell types and incubation for 24 h yielded maximal level of HPPH accumulation. The subcellular site of deposition is mitochondria/ER, which is clearly distinct from lysosomal compartment (Figure 3A, left 2 panels). The relative retention of HPPH during the subsequent 48 h chase period defined the cell type specificity (Figure 2).

Compared to HPPH, the erlotinib pyropheophorbide derivatives showed a 4-fold (PS 10, PS 19) to 20-fold (PS 16) lower initial uptake coupled with a reduced accumulation over the subsequent hours (Figure 2). The comparison also indicated the structure-dependent 48 h retention by the cell types with PS 10 as most prominently retained substrate for T-EC (Figure 2). Despite the lower uptake, erlotinib conjugates demonstrated a pattern of intracellular deposition equivalent to HPPH that, as shown for PS 10 and PS 11 in HN-143 T-EC (tongue cancer), is congruent with that of mitotracker (Figure 3A, right 2 panels). Comparable results have been obtained for the other erlotinib conjugates (except PS 27, which fails to diffuse into cells).

These co-culture experiments suggest that the erlotinib conjugates potentially encounter the intracellular domain of the plasma membrane EGFR during the internalization process. This encounter appears to be transient with the subsequent sequestration of the PSs at distant mitochondria/ER. Whether the inferred encounter after 30 min or after 24 h treatment is sufficient to alter EGFR kinase activity needed to be determined.

Indirect evidence for a failure of an interaction between erlotinib conjugates and EGFR as part of the uptake process was obtained when a maxilla cancer (HN-124) was identified whose tumor cells exhibited a rare phenotype of extremely low-level expression of EGFR protein and activity (Figure 3C). Despite of this deficiency, HN-124 T-EC did take up erlotinib-conjugates, such as PS 10, even more efficiently than HNT1 cells (Figure 3B) that express >100-fold higher level of EGFR (Figure 3C).

### 2.2. EGFR Activity in Cells Treated with Erlotinib Pyropheophorbide Conjugates

Diffusion of erlotinib-PS conjugates into cancer cells predicts the potential encounter of these with the cytoplasmic domain of EGFR and inhibition of its kinase function. To test this prediction, a cell-based assay of EGFR activity was applied (Figure 4). Cultured HN tumor cells with high level expression of EGFR protein and kinase activity, such as HNT1 cells, were pretreated with erlotinib or erlotinib-pyropheophorbides for 15 min in serum-free medium (Figure 4A,B) or for 24 h in medium containing 10% FBS (Figure 4C,D), followed by addition of EGF into the same medium and continued incubation for 15 min. Two pretreatment periods were chosen considering the different uptake kinetics and accumulation levels of the erlotinib-conjugates (Figure 2). EGF-dependent stimulation of EGFR kinase activity was evaluated by the autophosphorylation of EGFR and signal transduction toward the ERK1/2 pathway (Figure 4A). The inhibitory activities of non-conjugated erlotinib indicated an IC50 < 3 nM for P-EGFR and an IC50 ~5 nM for P-ERK1/2 (Figure 4B); values that are in the range of those determined in other cancer cell types [12,13].

Although there were substantial differences in the ability of HN cancer cells to take up erlotinib conjugates [11], there was no significant reduction of ligand-dependent EGFR activity, even after 24 h incubation (Figure 4C,D). One exception was PS 38, chlorin e6 with erlotinib attached to position 13 by a benzyloxy linker. PS 38 appreciably reduced the more sensitive read-out of EGFR activity, the autophosphorylation of the receptor protein (Figure 4D). The related compound PS 43, chlorin e6 with erlotinib attached to the same position 13 but via an ethylene glycol linker, was ineffective as EGFR inhibitor. The cell-based kinase assay was limited to a highest concentration of 3 μM PS, a concentration that exceeded several-fold that of the same PSs used for PDT.

### 2.3. PS-Mediated Photoreactions

Previous studies on the photosensitizing properties of pyropheophorbides, including HPPH, have indicated a tight correlation between the amount of PS taken up by the cells, as determined by the PS fluorescence, and the light-dependent photoreaction mediated by the PS [2]. The photoreaction was quantified by the covalent crosslinking of the latent transcription factor STAT3, loss of EGFR and activation of the stress p38 MAP kinase pathway. The same experimental analysis was applied to erlotinib conjugates. To assess the range of cell-specific PS action, T-EC preparations from different HN tumor samples with distinct binding activity for HPPH, were incubated with erlotinib-conjugates and the relationship of PS retention and response to the photoreaction determined.

In the example of an early passage of HN-138 T-EC (larynx tumor), which had indicated a low HPPH-retaining activity, the structurally related pair of PS 10 and PS 11 (differing in the presence of a methylester versus a carboxyl group at position 17^2^) was compared to HPPH and HPPH-ME (Figure 5A,B). Although the carboxylated compounds (HPPH and PS 11) were taken up at several-fold higher levels than the corresponding methylester derivatives (HPPH-ME and PS 10), after 72 h chase, only minor fraction remained (Figure 5A). In contrast, the methylester derivatives showed much lower egress rates with PS 10 as having the highest remaining level. Treatment of these cultures with therapeutic 665-nm light elicited photoreactions, which were proportional to the PS level (Figure 5B). As reported previously, the relative value for STAT3 dimerization in turn predicts the level of 24 h post-PDT survival of the cultures [2]. A survey of >100 separate monoytypic T-EC cultures for their lethal response to HPPH-mediated PDT indicated a “LD50” equivalent when ~1% of cellular STAT3 has been converted into dimers [2]. Complete lethal response was reached with 2–10% STAT3 dimerization. Hence, in the case of the HN-138 T-EC cultures used in Figure 5A,B, the effectiveness of PDT by the PSs is ranked PS 10 > HPPH-ME > HPPH > PS 11, with the last one yielding minimal, if any lethal reaction.

The application of erlotinib-PS in murine tumor models had indicated an unexpectedly high PDT efficacy for several conjugates. Although the cellular level of these PS did not appreciably exceed that of HPPH, higher lethal cell response and lower tumor recurrence were noted [11]. The standard analysis of PDT-mediated cell death in tissue culture was limited to the determination of cell viability 24 h after therapeutic light treatment. To assess PDT effect on long-term cell survival and recovery of proliferation, the post-PDT culture period must be extended to several days. Moreover, the influence of cellular interaction within the microenvironment of the tumor tissue needs to be considered. Indeed, growth analyses of many HN T-EC preparations have indicated that proliferation of the tumor cells is enhanced by stromal cell-derived growth factors. Hence, to characterize post-PDT tumor cell survival and growth, we chose to use reconstituted co-cultures of tumor epithelial and stromal cells as test system. The fact that stromal cells preferentially lost HPPH and erlotinib-conjugates (Figure 2), these cells were expected to survive PDT [14] and to potentially contribute growth-promoting factors to those tumor cells which escaped lethal PDT.

An example of an effective growth support activity by stromal cells was determined in combination with HN-143 tongue tumor cells (Figure 5C). Five-day-old co-cultures were incubated for 5 h with PS 5 or PS 10 and chased for 40 h in PS-free medium. Both PSs in the stromal cells and PS 10 in tumor cells were reduced below lethal PDT level, whereas PS 5 in tumor cells was retained at lethal level. Treatment of the co-cultures with 665-nm light followed by 5-day post-PDT incubation in full growth medium resulted, as expected, in the survival of the stromal cells, but with the elimination of all tumor cells (Figure 5C, right side panels). This seemingly paradox lethal action on PS 10-containing tumor cells, which should have survived at least in part, suggested an unexpected contribution of a PDT-dependent process.

### 2.4. Release of Erlotinib Activity by Photoreaction

Analysis of the photoreaction mediated by erlotinib conjugates in co-cultures in the presence of serum-containing medium was not amenable to yield conclusive information about growth-modifying components due to the complexity of reaction products. Therefore, we focused on determining the effect of the photoreaction on erlotinib conjugates alone. PS 10 dissolved in methanol was exposed to 660-nm light. Light treatment for 16 min (=5.3 J/cm^2^) led to ~40% reduction of PS fluorescence due to photobleaching. Aliquots of the reaction mixture were collected over the course of light treatment. The products were added to the cell-based EGFR kinase assay at a concentration equivalent to 3 μM original PS 10 (Figure 6A). A light treatment-dependent production of EGFR-inhibitory activity was detected. Maximal inhibitory activity was generated by 8 min light treatment (=2.65 J/cm^2^). Prolonged photoreaction led to a reduced recovery of EGFR-inhibitory activity, probably due destructive oxidative processes [15]. Using the densitometric values for the immunoblot signals for phosphorylated EGFR and ERK, a quantitative estimate of the photochemical release of erlotinib-type activity from PS 10 was obtained (Figure 6B). Three separate sets of photoreactions yielded comparable time course of inhibitor release, although the relative recovery of inhibitory activity was variable.

Based on testing serially diluted preparations of erlotinib and light-treated PS 10 (Figure 6C), we estimated a recovery of 1 to 10% of erlotinib-like inhibitory activity from PS 10. Equivalent recovery was obtained with PS 11, but insignificant amount with PS 5 and PS 8. Surprisingly, the photoreaction mediated by PS 27 generated the highest recovery of EGFR-inhibitory activity (Figure 6C), probably due to the presence of triple erlotinib moieties. We interpret the presence of EGFR-inhibitory activity in light-treated subset of erlotinib conjugates to be due to partial photochemical breakdown of the conjugates releasing erlotinib moieties now effective as EGFR kinase inhibitors. The precise molecular structure of the photoproducts in any of the PS preparations remains to be determined. The photoreaction with non-conjugated HPPH did not result in the production of detectable EGFR-inhibitory activity (Figure 6D) ruling out reactive components derived from the pyropheophorbide moiety alone.

### 2.5. Erlotinib PS Conjugate-Derived Photoproducts Attenuate Tumor Cell Proliferation

The biological relevance of PS-derived EGFR-inhibitory activity was evaluated using HN-143 T-EC cultures. These tongue cancer cells depend on EGFR signals to proliferate as demonstrated by the drastically reduced growth of HN-143 cells alone or in co-culture with HN tumor stromal cells in the presence of erlotinib (Figure 7A). Based on the EGFR-inhibitory activity present in light-treated PS 10 preparation (Figure 6A,B), the same products were applied to HN-143 T-EC cultures and compared to erlotinib (Figure 7B). After 3 days incubation, the lower cell numbers mirrored the EGFR-inhibitory profile. The cellular level of PS 10 detectable by fluorescence in the cultured cells was proportional to the fluorescence retained by PS 10 after light pretreatment. Four independent series of light-treated PS 10 samples confirmed the time course by which the photoreaction mediated the release of EGFR- and cell growth-inhibitory activity (Figure 6C and Figure 7C).

The same cell assay also highlighted the accuracy of testing erlotinib conjugate products by the response to light-treated PS 27 (Figure 7D). PS 27 has not been considered for further studies because it does not diffuse into cells, is poorly taken up by HN EC, and is primarily deposited into lysosomes. However, it mediates a photoreaction that releases erlotinib-equivalent activity that is now diffusible, inhibits EGFR activity and attenuates proliferation. Hence, despite its uptake into lysosomes, intact PS 27 could assist in reducing recovery of light-treated tumor cells through attenuation of EGFR signaling. The limiting factor would be the amount of PS 27, or of any other erlotinib conjugate, that can reach tumor tissue, and to be retained by cancer cells for mediating a photoreaction sufficiently effective for production of cell damaging reactive oxygen species [15] and for liberating erlotinib-related products that attenuate localized growth-supporting signaling. Notable is that most of the erlotinib-related activity should be confined to tumor cells. If cells survive the photoreaction, the PS-released erlotinib activity may exert EGFR inhibition and attenuated proliferation. It has already been shown that systemically administered erlotinib in combination with PDT, result in improved PDT outcome, despite the expected action of erlotinib in non-tumor cells [16,17,18].

### 2.6. In Vivo Distribution of Erlotinib Pyropheophorbides and Uptake by Tumor Tissue

The experimental work with cell culture models has provided information about preference of tumor epithelial cells for the various PSs (Figure 2), the site of intracellular retention (Figure 3) and the consequence of light-triggered photoreaction (Figure 5, Figure 6 and Figure 7). When attempting to translate this knowledge to the same tumor cell type but grown in vivo, one recognizes that the key information regarding systemic distribution and the level of uptake of the PSs at local tumor tissue needs to be gained through in situ measurements [8]. The subsequent assessment of the relationship between cellular concentration of PS in tumor tissue and immediate PDT response in vivo is expected to be much in line with what has been defined in ex vivo models.

The initial characterization of the erlotinib conjugates in murine models of epithelial tumor types had determined the organ distribution [11]. The method of choice was whole-body imaging of PS fluorescence of infused PS preparations. While these analyses had provided an estimate for the gross anatomical location of the PSs as a function of time following injection, the image resolution was insufficient to inform about cellular and subcellular level of each PS. To gain this information and to quantitatively compare retention of key erlotinib-PS conjugates, we resorted to the imaging of tissue cryosections. The same microscopic detection system was used that we have applied to the analysis of tissue cultures.

Patient-derived bladder cancer (BC-3) xenografts grown in NSG mice were chosen as test model (Figure 8). BC-3 tumor cells have high level expression of EGFR and form well-defined tumor cell clusters separated by stromal tissue within subcutaneously propagated xenografts. The in vivo properties of three structurally related PSs, PS-531, PS 10 and PS-908, were compared. PS-531 [6] had served as substrate for the synthesis of PS 10. PS-908 [10] tested an alternative position of erlotinib in that erlotinib was conjugated via a benzyl linker to position 17^2^ as found in PS 8. Organ distribution of the PSs was determined 24 h after intravenous injection by fluorescence microscopy of 10 μm cryosections. All three PSs show uniform and high-level retention in liver with PS-908 reaching the highest concentration. Kidney retained the same PSs in a more heterogenous pattern within tubular cells. BC-3 xenograft on average had level equal to 25% of the liver for PS-531 and PS 10, and 10% of the liver for PS-908. The most relevant information gained by the in vivo experiments as shown in Figure 8 is that the erlotinib conjugates reached within 24 h a fairly uniform access to the tumor cell clusters, were retained by the tumor cells markedly above the stromal area, and presented subcellular deposition as already determined of tissue culture cells. Few regions within large tumor area showed a reduced level of PS fluorescence at the center and elevated level at periphery, suggesting heterogeneity in intra-tumoral distribution or diffusion of the PSs.

A separate experiment tested the regional distribution of PS 10 in a subcutaneous HNT1 xenograft by using the same treatment and analysis technique as applied for BC3 xenograft-bearing mice. The cryosection of the HNT1 tissue indicated an intra-tumoral distribution of PS 10 with heterogeneity among tumor cell clusters, which was comparable to that seen for BC3 (Appendix A).

The level of PS fluorescence detected in cryosection allows a prediction about the relative effectiveness of the photoreaction and its impact on cell survival. In the case of PS 10 treatment, the release of EGFR-inhibitory activity as a function of the local concentration of PS 10 may appreciably contribute to the post-PDT programing of the surviving tumor cells, activity of the local stromal cells and of inflammatory cells entering post-PDT site. This suggested mechanism may also explain the higher PDT efficacy of PS 10 compared to HPPH detected in the previous in vivo tumor treatment models [11].

## 3. Materials and Methods

### 3.1. PS Preparations

Synthesis, isolation and analysis of HPPH and its 17^2^ methylester derivative HPPH-ME [19], PS-531-COOH and PS-531-ME [19], erlotinib-conjugates PS-908 [10], and PS 5 to PS 43 (Figure 1) [11] have been previously reported. Each PS preparation used for tissue culture and in vivo studies has been verified by mass spectrometry and nuclear magnetic resonance analysis for purity of >96% and to be without trace of free erlotinib. Stock solutions of the PSs with concentrations ranging from 0.3 to 1.2 mM were prepared in H_2_O containing 5% glucose, 0.1% Tween 80 and 0.1% ethanol and diluted in culture medium or injection solution immediately prior to use. The concentrations of PSs in the treatment media were confirmed by using medium aliquots, diluted 10-fold in methanol, removed precipitated proteins by centrifugation and determined PS concentration in the supernatant by fluorometry. Stock solution of erlotinib (Cell Signaling Technology, Danvers, MA, USA) was prepared in DMSO (10 mM) and aliquots diluted into medium for cell treatment.

### 3.2. Light-Treatment of PSs

PSs were dissolved in methanol yielding 3 μM solutions. Aliquots of 2 mL were added to wells in 6-well cluster plates and exposed at room temperature to 660 nm light (dye laser) of a 15-cm diameter spot size at 5.5 mW/cm^2^ for a total of 16 min (maximal fluence of 5.3 J/cm^2^). Samples were dried in a speed vacuum centrifuge (Savant) and residues dissolved in 2 μL DMSO followed by dilution into 2 mL RPMI containing 10% FBS. Aliquots of these samples were used either to treat HNT1 cells to determine inhibition of EGFR kinase activity or to HN-143 T-EC cultures for inhibiting proliferation.

### 3.3. Tissue Cultures

Primary cultures of tumor and stromal cells were generated from surgical specimens prepared by the Pathology Department and distributed by the Tissue Procurement Service at the RPCCC by the techniques described previously [2,20]. The use of the patient-derived samples was approved by IRB. Selective growth of epithelial cells, under suppression of fibroblasts, was achieved by culturing in serum-free keratinocyte medium supplemented with recombinant EGF and cholera toxin. Long-term cultures of head and neck tumor epithelial cells (HN T-EC) were generated by adapting the cells to grow in RPMI containing 10% FBS. Primary stromal cell cultures were maintained in DMEM containing 10% FBS. To identify cell type-specific retention of PSs, stromal cells were stained prior to use by treatment with CFSE (Abcam, Cambridge, UK). Co-cultures were generated by combining T-EC with tumor fibroblasts (T-Fb) at ratios ranging from 1:10 to 1:100, plated onto collagen-1 matrix, and maintained in RPMI containing 10% FBS.

### 3.4. Xenografts

Tissue aliquots of patient-derived bladder cancer (BC-3) xenograft were passaged by subcutaneously implantation into the upper side of the hind leg flank of six-week old NSG mice (NOD.Cg-*Prkdc^scid^ Il2rg^tm1Wjl^*/SzJ; RPCCC Tumor Model Service). HNT1 cells derived from human tongue cancer xenograft were passaged by subcutaneous injection of 1 × 10^6^ cells in 100 μL PBS into the hind leg of eight-week-old SCID mice (CB-lgh-lb/lcrTac-PrMscid/Ros; Department of Laboratory Animal Resources, RPCCC). When palpable xenografts reached ~5 mm diameter, PS (3 μmol/kg) was injected intravenously and the animals were euthanized 24 h later. Liver, kidney, and tumor were collected. Organ pieces were immediately frozen in liquid nitrogen, combined, and embedded in a single block of optimal cutting temperature compound (OCT). Cryosections (10 μm) were generated and cellular level of PS was determined by florescence microscopy. Cryosections were fixed in formalin and stained with eosin and hematoxylin. All animal work was approved by IACUC.

### 3.5. Cellular Uptake of PS

Cultures of tumor cells, alone or in co-culture with tumor stromal cells, were incubated with RPMI containing 10% FBS and 1.6 to 3 μM PS at 37 °C for 30 min to 24 h (=“uptake”. Depending on the experimental setting, PS treated cells were maintained for additional time periods in PS-free medium (=“chase”). Subcellular deposition and level of PS were visualized by fluorescence microscopy (Zeiss Axiovert). Fluorescence was recorded in monochrome format and quantified using image scanning (ImageQuant; Amersham/GE Healthcare, Chicago, IL, USA). The net pixel values were related to image exposure time (within linear range of signal detection), image size and cell numbers and were expressed in arbitrary fluorescent units (FU) [2]. For optimal presentation of fluorescence pattern in the figures, the fluorescence signals are reproduced in monochrome (black and white). Lysosomal or mitochondrial localization was defined by staining with Lysotracker Green or Mitotracker Red (ThermoFisher, Grand Island, NY, USA), retrospectively. For image presentation, mitrotracker fluorescence has been false colored in green and PS fluorescence in red.

### 3.6. EGFR Kinase Inhibition Assay

Depending upon the experimental setting (see figures), confluent monolayers of HNT1 cells or HT-143 T-EC in 24-well cluster plates were pre-incubated with serum-free RPMI or RPMI containing 10% FBS, with erlotinib or erlotinib-conjugates (0–3000 nM), for 15 min or 24 h at 37 °C. EGF (100 ng/mL) was added to the cultures and incubation continued for 15 min. Cells were lysed within the culture wells and aliquots used to analyze for proteins by Western blotting.

### 3.7. Cell Growth Inhibition by Erlotinib

HN-143 T-EC [2] were plated into 6-well cluster plates (1 × 10^5^ cells/well) atop a collagen-1 matrix in 2 mL RPMI containing 10%FBS, with or without erlotinib or erlotinib-PS conjugates. Cultures were incubated for 3–4 days. The cell density in each well immediately after plating and at the end of culture period was imaged and the number of viable cells counted after release of the adherent cells by trypsin digestion. In co-culture setting, HN-143 T-EC were combined with HN T-Fb (ratio 1:10) and plated into 6-well collagen-1-coated cluster plates in 2 mL RPMI containing 10% FBS. After 3–5 day incubation, allowing formation of tumor cell clusters within stromal cell layer, fresh medium with erlotinib or erlotinib-PS conjugates (0–3000 nM) was added and cultured for 3–5 days. Growth of tumor cell clusters was followed by microscopic imaging.

### 3.8. Photoreaction

Cell cultures were exposed to 665 nm light (diode laser) for 9 min at 5.5 mW/cm^2^ (=3 J/cm^2^) in RPMI containing 10% FBS in a 37 °C tissue culture incubator. The cells were lysed in situ in RIPA buffer and aliquots containing 10 μg protein were analyzed by SDS-PAGE and immunoblotting for STAT3, EGFR, and signaling proteins [21]. Crosslinking of STAT3 was quantified as described previously [22]. Cell survival was determined in cultures 24 h after light treatment by counting cells using trypan blue exclusion. Cultures that were exposed to light, but without PS, served as a reference for 100% survival.

## 4. Conclusions

The functional analyses of erlotinib-pheophorbide conjugates demonstrated that the presence of the tetrapyrrole moiety prevents an effective interaction of the erlotinib moiety with the cytoplasmic receptor domain involved in kinase activity. Despite the predicted correct structural presentation of the bound erlotinib to the ATP binding site, the dominant lipid nature and preference for membrane association of the pheophorbide moiety seemingly prevents this to occur. Hence, the originally envisioned targeting function of erlotinib to direct a photosensitizer to EGFR-expressing cells is not observed. However, the results of the study of tumor cell type-specific retention indicates that erlotinib, separate to its EGFR-binding property, also represents a structural element that in combination with the attached pheophorbide gained a favorable binding activity to the mitochondria/ER components. The susceptibility of some of the erlotinib conjugates to react to the light-activated oxidative process by a degrading process deliberating product(s) with erlotinib properties indicates a non-anticipated benefit for PDT of EGFR signal-dependent tumor. The localized action of these products, within the timeframe of its effective retention in the microenvironment of the tumor tissue, will moderate EGFR-directed signaling in all resident cell types. Some of these activities are thought to be in part responsible for the heightened PDT action on tumors in vivo.

## Figures and Tables

**Figure 1 ijms-23-11081-f001:**
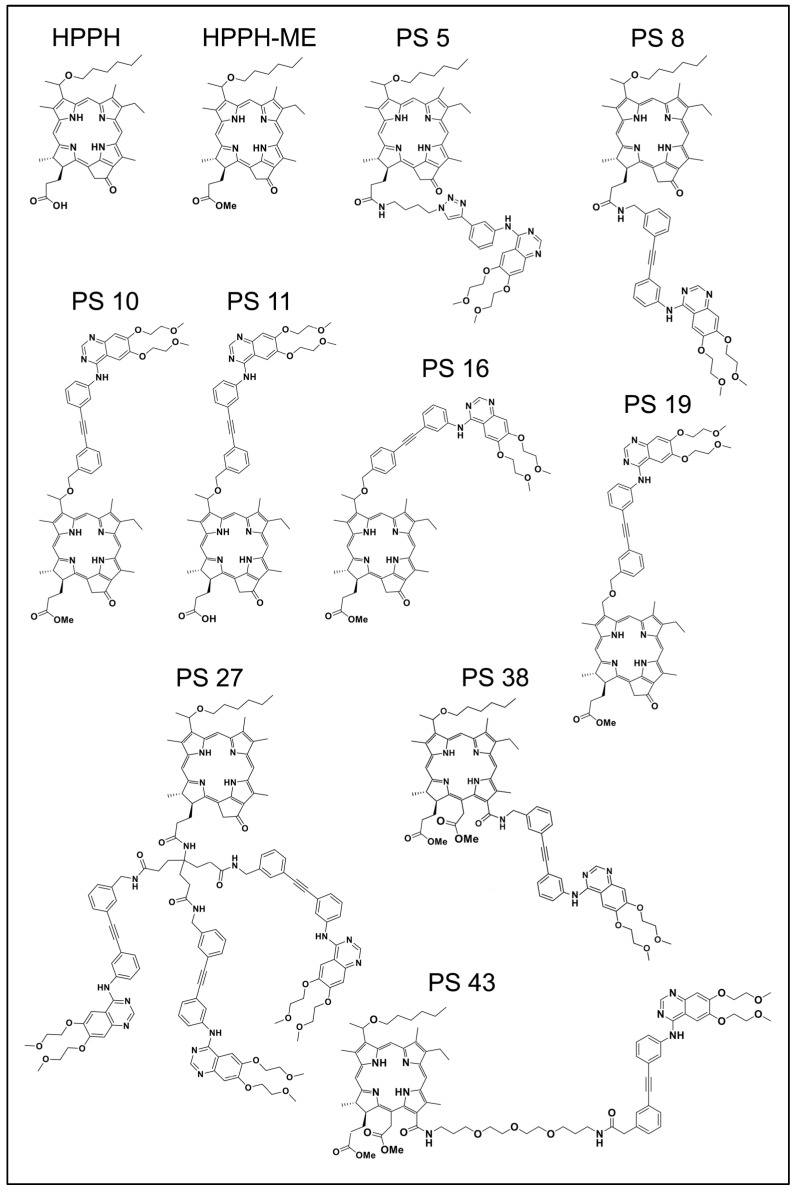
Structures of the erlotinib chlorin conjugates used in this study.

**Figure 2 ijms-23-11081-f002:**
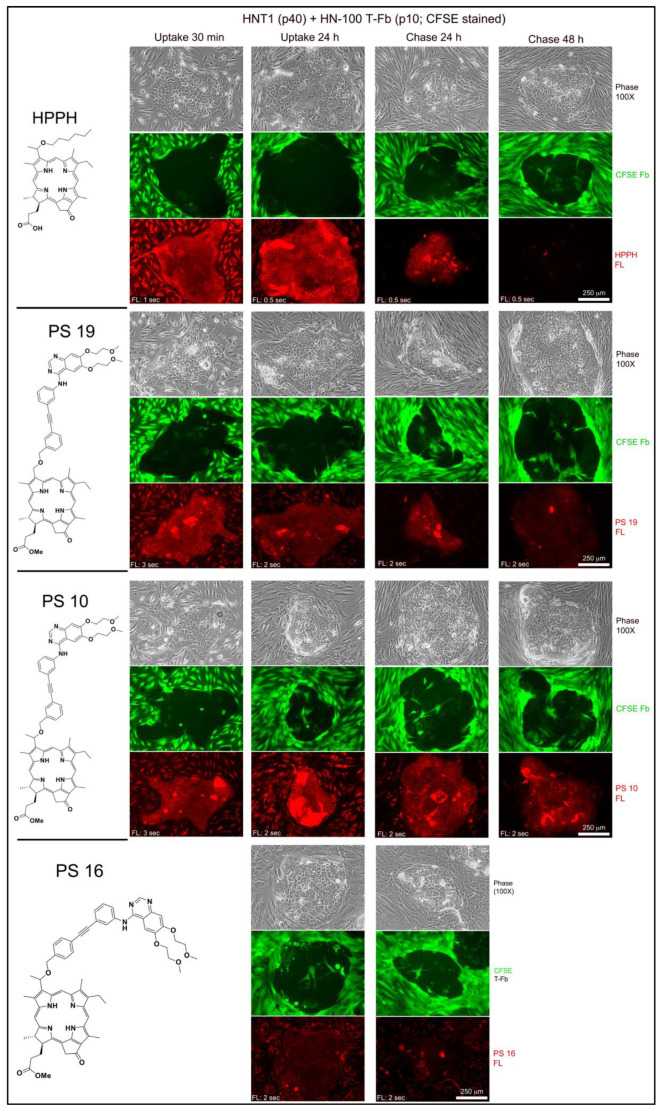
Uptake and retention kinetics for HPPH, PS 19, PS 10, and PS 16 in co-culture of HNT1 and CFSE-stained HN-T-Fb. Three-day old co-cultures of HNT1 (p 40) and CFSE-stained HN-100 T-Fb (p 10) were incubated in DMEM containing 10% FBS and 3 μM of the PS indicated at the left. Initial binding and uptake were determined after 30 min treatment at 37 °C and maximal uptake after 24 h. Retention of the PSs was assessed after 24 and 48 h chase periods. Note, the difference in the length of exposure for PS fluorescence.

**Figure 3 ijms-23-11081-f003:**
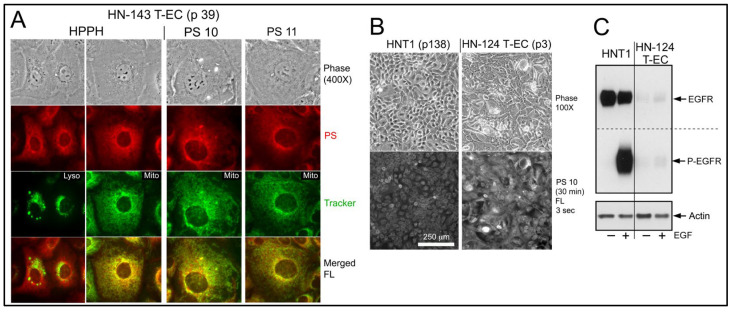
Subcellular location of HPPH, PS 10 and PS 11 and relationship of EGFR expression with uptake of PS 10. (**A**) Subcellular site of HPPH, PS 10 and PS 11 retention was determined in HN-143 T-EC cells after 24 h uptake (3 μM) and 24 h chase. Mitotracker served as marker for mitochondia. To demonstrate the distinct pattern of lysosomal compartment, HPPH-treated cells were also stained with lysotracker (left most panel). (**B**) Uptake of PS 10 (3 μM) by HNT1 cells and HN-124 T-EC (p 3) cultures. (**C**) Difference in the expression of EGFR protein and kinase activity (15 min treatment with EGF) by HNT1 and HN-124 T-EC.

**Figure 4 ijms-23-11081-f004:**
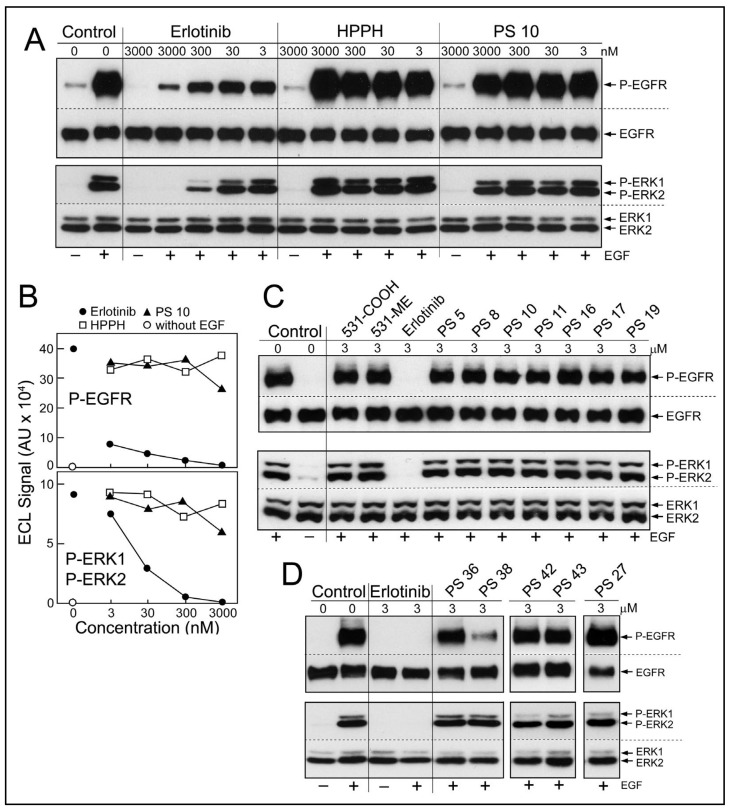
Effect of erlotinib and erlotinib conjugates on EGFR kinase activity. (**A**) HNT1 cells (p 186) treated for 15 min in serum-free media with increasing concentration of erlotinib and the PSs indicated at the top, followed by 15 min incubation with EGF (100 ng/mL). The cells were immediately lysed and aliquots of the cell extracts analyzed by Western blotting for the indicated proteins. (**B**) The immunodetectable signals (enhanced chemiluminescence, ECL) of the bands representing phosphorylated EGFR and ERK1/2 in “A” were quantified. (**C**,**D**), In two separate sets of experiments, HNT1 cells (p 163 and p 164) were treated for 24 h with media containing 10% FBS and the indicated compounds listed at the top. The function of the EGFR was assessed by treatment for 15 min with EGF added to the same media. The levels of phosphorylated EGFR and ERK1/2 were determined by immunoblot analysis. PS 36 is the substrate to which erlotinib has been conjugated yielding PS 38, and PS 42 is the substrate for synthesis of PS 43.

**Figure 5 ijms-23-11081-f005:**
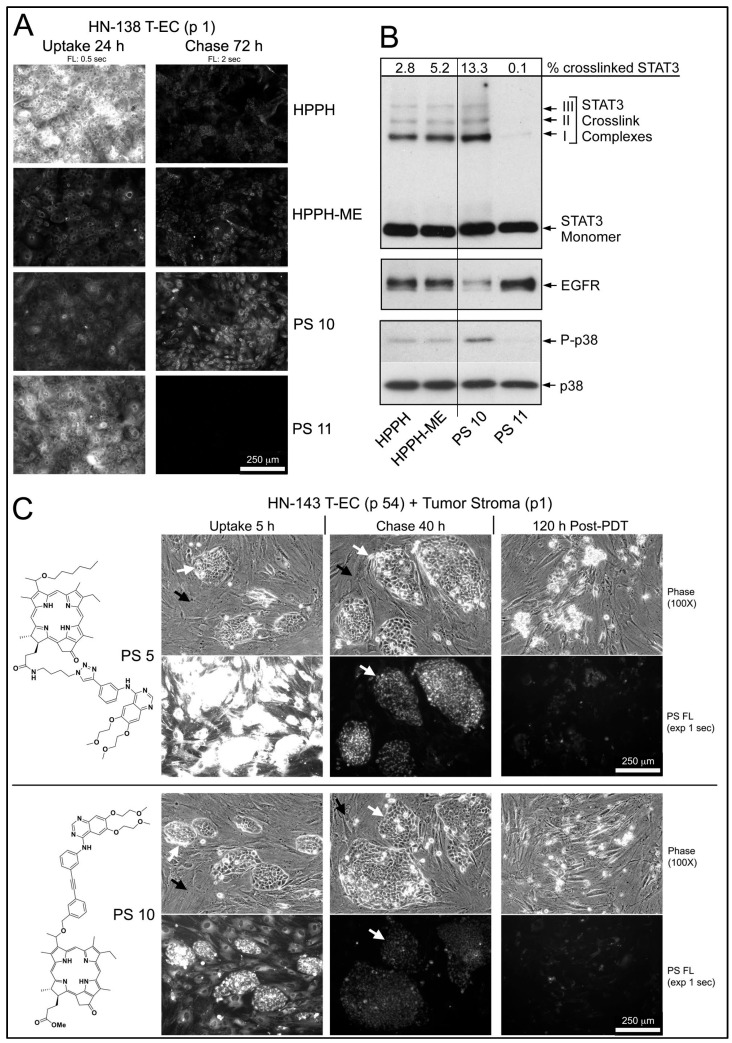
Photoreaction and PDT response by HN T-EC. (**A**), Cultures of HN-138 T-EC (p 1) were incubated for 24 h in media containing 10% FBS and 1.6 μM of the indicated PSs. The cells were then cultured for 72 h in PS-free media. (**A**), Cell-associated fluorescence was imaged by microcopy. (**B**), Light-mediated photoreaction of the cells in (**A**) (9 min, 5.5 mW/cm^2^, 3 J/cm^2^) resulted in products of photoreactions that are tightly correlated with PS level. (**C**), Co-culture of HN-143 T-EC (p 54) and T-Fb (p 1) were treated for 5 h with RPMI containing 10% FBS and 1.6 μM PS 5 or PS 10, followed by 40 h chase. The cultures were exposed to 665 nm light (3 J/cm^2^) and maintained in growth medium for additional 120 h. Culture morphology and level of PS fluorescence were imaged by microscopy using identical settings. Stromal cells survived but all tumor cells were killed, some of which are still attached as dead cell aggregate to the support matrix. The morphologically identifiable tumor cell clusters are indicated by white arrows and interspersed stromal cell layer by black arrows.

**Figure 6 ijms-23-11081-f006:**
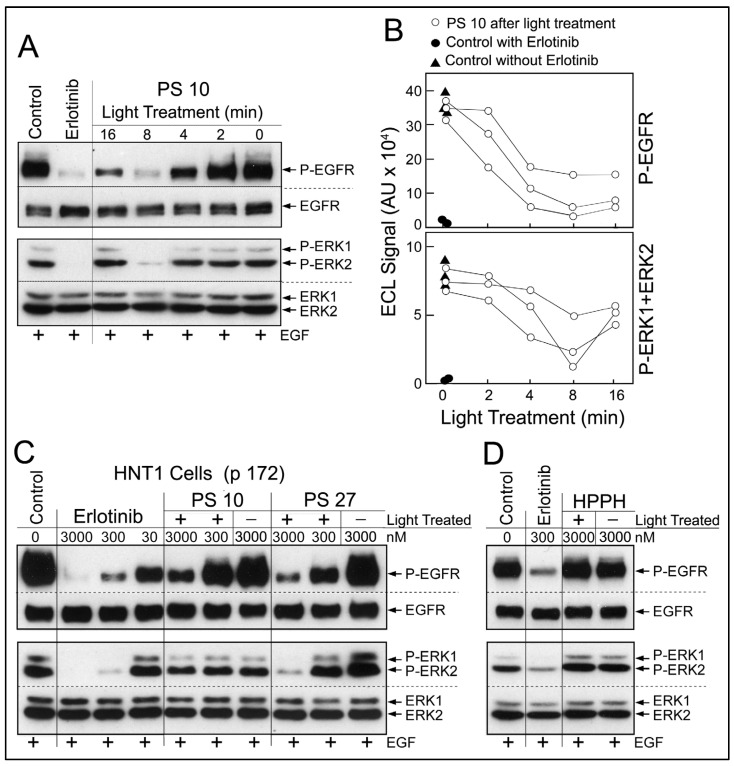
Recovery of EGFR-inhibitory activity following light treatment of PS. (**A**), Solution of PS 10 (3 μM in methanol) was treated with 660 nm light for 16 min (0–5.3 J/cm^2^). Samples collected over the course of the light treatment were dried. Residues were dissolved in DMSO and diluted (1:1000) in DMEM containing 10% FBS yielding a concentration equivalent of 3 μM original PS 10. These medium samples were subjected to a 30 min EGFR kinase inhibition assay using HNT1 cells (p 170). The activation of EGFR signaling by 15 min treatment with EGF was identified by immunoblotting. (**B**), Three separate sets of light treatment of PS 10 were analyzed for the recovery of EGFR kinase-inhibitory activity by quantifying the immunoblot signals for phosphorylated products. (**C**), Ten-fold diluted samples of 8 min light-treated (2.65 J/cm^2^) PS 10 and PS 27 were used to measure inhibition of EGFR kinase activity in HNT1 cells (p 172). (**D**), As control, HPPH was subjected to 8 min light treatment (2.65 J/cm^2^) and products tested for effects on EGFR kinase activity in HNT1 cells (p 173). Note: images of separate immunoblots have been integrated into the shown composite figures.

**Figure 7 ijms-23-11081-f007:**
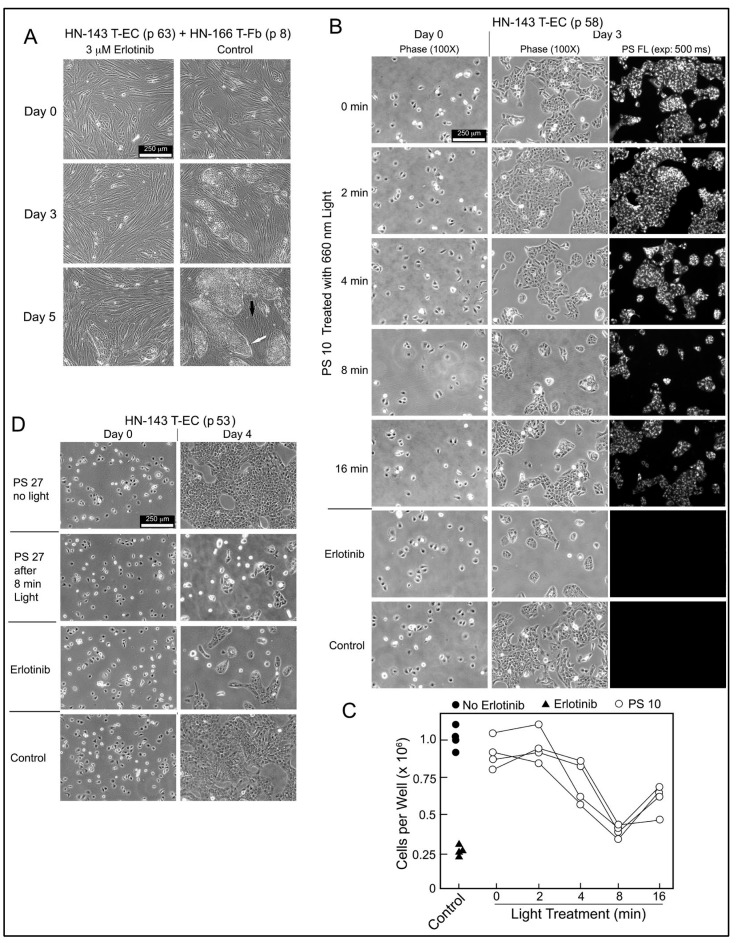
Inhibition of tumor cell growth by erlotinib and light-treated erlotinib-conjugates. (**A**), Three-day old co-culture of HN-143 T-EC (p 63) and HN-166 T-Fb (p 8) were incubated for 5 days in RPMI containing 10% FBS and with or without 3 μM erlotinib. Growth of the tumor cell clusters was monitored by microscopy. Tumor cell cluster is indicated by white arrow and interspersed stromal cell layer by black arrow. (**B**), HN-143 T-EC (p 58) were plated at 1/20 dilution (1 × 10^5^ cells per well in 6-well plate) in RPMI containing 10% FBS and 3 μM aliquots of a PS 10 preparation illuminated between 0 and 16 min (0–5.3 J/cm^2^). As reference served one well that included 3 μM erlotinib. Cells and PS fluorescence were imaged after 3-day incubation. (**C**), Four independent series of light treatments of PS 10 were tested for the effect on cell growth as shown in (**B**). The numbers of viable cells recovered after 3 d incubation were determined. (**D**), The erlotinib-like activity of 8 min light-treated PS 27 was tested on cultures of HN-143 T-EC (p 22) with incubation carried out for 4 d.

**Figure 8 ijms-23-11081-f008:**
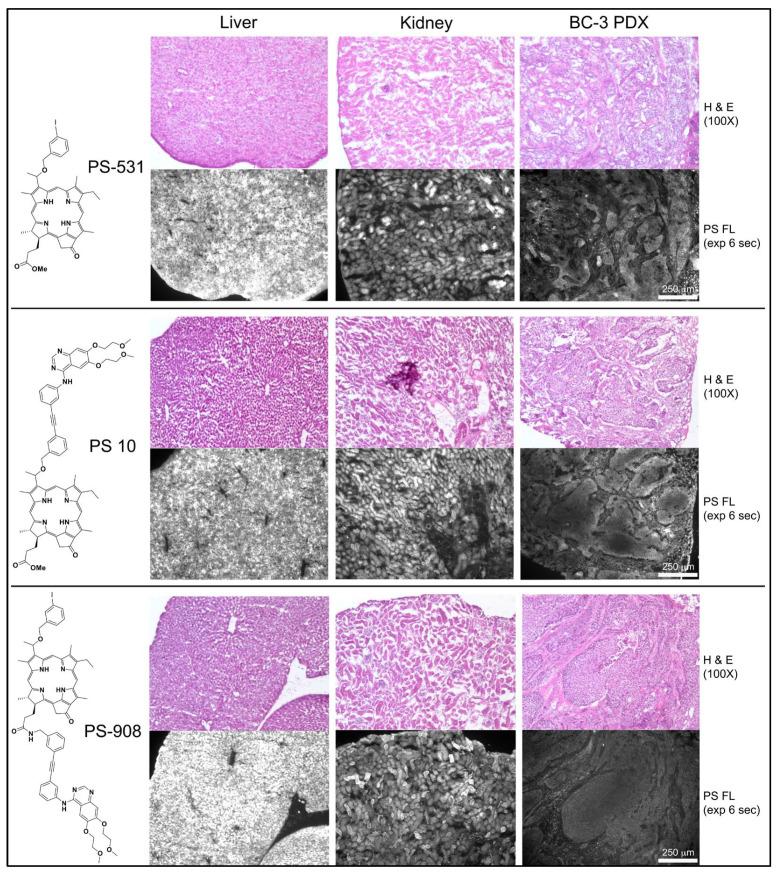
In vivo distribution and localization of PS in bladder cancer PDX bearing mice. NSG mice bearing patient-derived bladder cancer (BC-3) xenografts received an intravenous injection of PSs (3 μmol/kg) and organs were collected 24 h later. Cryosections (10 μm) were imaged for PS fluorescence and compared to the H&E-stained histology.

## Data Availability

All experimental data were generated in the authors’ laboratories and specific information will be made available by the authors upon request.

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
