# Peer review of "Tumor Cell-Specific Retention and Photodynamic Action of Erlotinib-Pyropheophorbide Conjugates"

_ijms, 2022, doi:10.3390/ijms231911081_

Round 1

Reviewer 1 Report

The authors aimed to define the efficacy of different conjugates (erlotinib + PSs) to modulate EGFR functions and tumor growth. Unfortunately, it is not clear to ultimate goal of this manuscript. The manuscript is well written and has a good potential but cannot be accept in the present form.
Why would be a benefit to use these conjugates instead of erlotinib alone? If one of the aims is to evaluate the effect of tumor growth, I think would be necessary add several experiments about the modulation of several proliferation and apoptosis markers.
I wondered why the light-treatments of PSs were performed on the isolated molecules and then incubated with cells instead of the   cells incubation with PSs before and then  the light treatment.
Is it possible to synthetize these conjugates using other TKIs?
Do the authors know the mutation status of EGFR? This information could be interesting to determine the efficacy of erlotinib treatment and to use the appropriate compound concentration.
In the text the authors never described the acronym PDT.

Reviewer 2 Report

The work presented by Erin C. Tracy et al developed tumor cell selective erlotinib-pyropheophorbide Conjugates photosensitizer for simultaneous imaging tracking and targeting.

Strategies for inducing tumor cell-specific properties and simultaneously eliciting PDT effects by utilizing selectivity for EGFR are thought to be well established.

The direction and means of experiments to prove hypothesis were also well presented. Unfortunately, unlike the initial goal, the tendency to lower EGFR activity was not clear, and it seems that the direction of the experiment was focused on showing the residual ability of PS.

Nevertheless, if the therapeutic effect on cancer cells is large, it can be said to have academic and medical significance.

1. It would be better for readers to add data expressing the numerical values ​​for fluorescence shown in Fig.2.

2. In Fig. 3, the data showing the selectivity for mitochondria is shown with PS11. In this study, it is better to show the data using PS (eg PS10), which is considered to have the best performance. Also, it would be good to present the overlapping constant value (pearson's' correlation coefficient) for the co-localization image.

3. It would be good to mark the stromal cells and tumor cells with arrows to make it easier for readers to distinguish them. And above all, by showing the therapeutic effect according to the concentration of PS and the light irradiation time and intensity for PDT together, there is a need to suggest whether appropriate conditions are used.

4. Why haven't there been any attempts to see the effects of PDT in the organization? And, in order to present the potential as a therapeutic agent through residual, it would be good to show the signal of the image at least as a qualitative or quantitative value in the ROI.

5. Scale bar is missing in all images. It would be nice to add.

Round 2

Reviewer 1 Report

I am satisfied with the changes made